# STEP IN: Supporting Together Exercise and Play and Improving Nutrition; a Feasibility Study of Parent-Led Group Sessions and Fitness Trackers to Improve Family Healthy Lifestyle Behaviors in a Low-Income, Predominantly Black Population

**DOI:** 10.3390/ijerph20095686

**Published:** 2023-04-28

**Authors:** Michelle C. Gorecki, Megan E. Piotrowski, Courtney M. Brown, Radhika R. Teli, Zana Percy, Laura Lane, Christopher F. Bolling, Robert M. Siegel, Kristen A. Copeland

**Affiliations:** 1Division of General and Community Pediatrics, Cincinnati Children’s Hospital Medical Center, 3333 Burnet Avenue, MLC 7035, Cincinnati, OH 45229, USA; 2Department of Pediatrics, Division of Primary Care Pediatrics, Nationwide Children’s Hospital, The Ohio State University College of Medicine, Columbus, OH 43205, USA; 3Department of Pediatrics, New York University Grossman School of Medicine, New York, NY 10016, USA; 4Department of Pediatrics, University of Cincinnati College of Medicine, Cincinnati, OH 45267, USA; 5Division of Biostatistics and Epidemiology, Cincinnati Children’s Hospital Medical Center, Cincinnati, OH 45229, USA; 6The Heart Institute, Cincinnati Children’s Hospital Medical Center, Cincinnati, OH 45229, USA

**Keywords:** obesity management, pediatric obesity, group weight management, peer support, peer-led group sessions, primary care pediatrics, patient engagement, health disparities

## Abstract

Background: Pediatric obesity is prevalent and challenging to treat. Although family-centered behavioral management is the gold standard, many families face structural inequities to its access and efficacy. Identifying ways to manage pediatric obesity within primary care is needed. Methods: This feasibility study included three sequential trials of peer-led group sessions occurring biweekly or monthly between 3/2016 and 2/2017. Parent–child dyads were recruited from a large academic primary care clinic via mailed invitations, prioritizing patients living in local zip codes of historical disinvestment. Eligible patients were 6 to 12 years with a body mass index ≥85th percentile, with parent and child interest in making healthy lifestyle changes, and English speaking. Results: 27 dyads participated, 77% were non-Hispanic Black. Retention and attendance rates were highest in the initial four-session biweekly pilot (100%, 0 dropouts), high in the full six-session biweekly cohort (83%, 1 dropout), and moderate in the monthly cohort (62.7%, 4 dropouts). Families reported high satisfaction with the sessions (4.75/5). Qualitative comments suggested social connections had motivated behavior change in some families. Conclusion: Parent-led group sessions for pediatric weight management show promise in engaging families. A future large trial is needed to assess behavior change and anthropometric outcomes.

## 1. Introduction

Obesity is a highly prevalent condition of childhood (18% of all US 6–11-year-olds) with significant racial/ethnic disparities (29% of Black children, 23% of Hispanic children) [1] associated with an increased risk of multiple adverse effects—depressed mood and academic achievement, diabetes, stroke, and premature death [2,3,4]. Children and adults with higher weight experience significant weight stigma which leads to negative social, psychological, and physical health effects [5]. Obesity’s root causes are multifactorial (including upstream contributors of toxic stress such as adverse childhood experiences and the experience of racism), making it particularly challenging to treat or prevent [6]. However, there are modifiable risk factors that involve changing two essential health behaviors: diet and exercise [7]. These behaviors lie squarely within the domain of pediatrics and preventive health. Thus, public health experts have called for innovations in primary care to address them [4,8,9,10,11]. Family-centered behavioral treatment is the gold standard for effective treatment of childhood obesity in public health settings. Unfortunately, most interventions have been designed for and tested in well-educated, non-Hispanic White families [12,13]. There are challenges with recruiting research participants with low incomes and of racial/ethnic minority backgrounds, with good reason given the historical exploitation of many groups in medical research [14]. There is critical need for culturally tailored family-based interventions, i.e., interventions that resonate with and work well for families of color as well as for those with limited resources such as time, money, and access to safe places for physical activity. Like many cities, our local community has a disgraceful history of redlining practices, with neighborhoods of predominantly Black residents facing decades of disinvestment, exacerbating racial inequities in access to safe outdoor spaces and healthful foods [15,16]. Black youth also face disproportionate marketing of unhealthy foods [17]. Furthermore, effective means within primary care to prevent and treat obesity, unhealthy diet, or inactivity are lacking [4], and several research interventions have had only modest effects, even with one-on-one family-centered counseling [18,19,20].

Within the context of a 20-min well-child check, and with competing patient-related medical and behavioral concerns, it is difficult to counsel effectively about a patient’s diet or to effectively promote physical activity [21]. To meet the recommended standard of care, the two options for primary care clinicians are to bring the patient back for a one-month follow-up visit to check progress on diet and exercise or to refer to an outside intensive weight management program [4]. Although an external comprehensive weight management program is appropriate and effective for some, the demanding program may be too intensive or burdensome for many low-income families with limited transportation and time off work to make appointments. Furthermore, both options have poor follow-up. A retrospective chart review in our large clinic, serving a population who are primarily low-income and African-American, indicates poor follow-through with either option: only 13% of patients return for the follow-up weight check within primary care, and only 36% of patients who are referred to our comprehensive weight management clinic make it to an initial appointment within 6 months [22]. Of those who do make it to an initial comprehensive weight management program appointment, less than half continue beyond the initial visit [23]. Of those continuing treatment in a comprehensive weight management clinic over one year, only one third of patients have a meaningful decrease in body mass index (BMI) percent of the 95^th^ percentile associated with a decrease in cardiometabolic risk [24].

What primary care lacks is a system to activate patients to make healthy changes in diet and exercise and to support patients when they inevitably encounter barriers, particular low-income families that experience food insecurity and lack safe places to be active. The current model evokes low patient engagement and uptake in both primary care and in tertiary care approaches. Furthermore, there is no opportunity for patients/peers from the same neighborhood and/or from similar financial and cultural circumstances to share “practical tips” to overcome barriers [25] by imparting relevant and neighborhood-specific knowledge about good places to purchase fresh fruits/vegetables and safe parks to visit, or tips about making meals for picky eaters on a tight budget as a single parent. Peer engagement is one strategy to promote patient activation and empowerment [26].

The purpose of this trial was to assess the feasibility and acceptability, within a primary care clinic population which is predominately Medicaid-insured and identifies as non-Hispanic Black, of a group dietary and activity-promoting intervention for patients and their caregivers. Our objective in the Supporting Together Exercise and Play and Improving Nutrition (STEP IN) feasibility study was as follows: (1) Develop and pilot intervention content; (2) Determine the feasibility and acceptability of group nutrition sessions and FitBit intervention; (3) Assess participation and retention; (4) Optimize the content of the intervention through real-time monitoring of children’s and parents’ reported diet and screen time data, as well as iterative changes based on participant feedback. The intervention consisted of the following: (1) six parent-led group sessions focusing on evidence-based and practical tips for healthful eating and physical activity, (2) distribution of fitness trackers (Fitbit Flex^®^, Fitbit LLC, San Francisco, California, USA) to parent–child dyads, and (3) continuous monitoring of dietary intake through brief pulse surveys, steps with Fitabase software, and display of group-level data to participants at group sessions. The intervention was conceived and designed with formative input from parents of patients from our clinic and from a team of experts in pediatrics, obesity, improvement science, and psychology.

## 2. Materials and Methods

### 2.1. Trial Design

This feasibility study, conducted between March 2016 and February 2017, consisted of three non-randomized sequential trials with a single intervention arm: (1) a two-month initial pilot session with four sessions occurring every other week (biweekly) to practice curricular programming, logistics, and administration of study measures, (2) a three-month session with six sessions occurring biweekly, and a (3) a six-month session with six monthly sessions. The rationale for this pilot session was to test logistics prior to implementation of the full curriculum in the biweekly and monthly cohorts. The two groups with six sessions used all procedures intended for a future full trial. This design was chosen to allow comparison of acceptability and retention results between the biweekly group and the monthly group. The study was approved by Cincinnati Children’s Hospital Medical Center Institutional Review Board. The STEP IN trial was registered at the US National Institutes of Health (ClinicalTrials.gov (accessed on 22 February 2023)) #NCT02724839.

### 2.2. Formative Input of Intervention Design

Study staff conducted four focus groups with 37 parents/caregivers of children with overweight/obesity to solicit feedback on the proposed intervention design and suggested practical tips. Focus groups were moderated by an experienced moderator (Dr. Alisa Balestra), videotaped, and attended by two study team members (K.A.C. and Z.P.). The goal was to understand what nutritional coaching families wanted or needed (e.g., meal planning, tips for purchasing inexpensive healthy foods, tips for increasing vegetable consumption); how they wanted it delivered (e.g., didactic versus facilitated group discussion); whom they would like to deliver the content (e.g., health professional, community health worker, peer); and where they would want it delivered (e.g., at the clinic versus a community setting). Examples of group-based nutritional coaching and peer support programs were presented, and families were asked to react and suggest changes. Focus group participants also completed written surveys. Interviews and focus groups used user-centered design principles to understand what people think and feel about a new proposed process or design, rather than qualitative methodologies designed to understand beliefs and behaviors [27,28,29]. Marketing-style focus groups sought input on the proposed study as well inquiring about barriers and successes with healthy meals and encouraging physical activity for their children. Videos and notes from the focus groups were reviewed by four study team members (Z.P., R.R.T., M.E.P., and K.A.C.). Two consensus notes documents were created and shared with the design team advisory board. Many focus group participants were interested in participating in STEP IN after being a part of the focus groups.

A design team advisory board was formed to create and provide on-going input on the intervention. The design team advisory board was interdisciplinary, with relevant expertise consisting of a pediatrician and nutrition/activity researcher (K.A.C.), a pediatrician and quality improvement expert (CMB), a pediatrician and director of weight management clinic (R.M.S.), a community pediatrician and childhood obesity researcher (C.F.B.), a psychologist (Dr. Monica Mitchell), and five parents who had participated in focus groups and indicated interest in helping with the study. The design team advisory board met eight times from September 2015 to April 2017 to design and refine the intervention.

### 2.3. Participants and Recruitment

Parent–child dyads were recruited from a large academic pediatric primary care clinic whose patient population is primarily (70%) Black and Medicaid-insured (90%). Child participants were eligible for participation if they were between the ages of 6 and 12 years, their BMI was greater than the 85^th^ percentile for age and sex [30], and if their identified primary medical home was the primary care clinic based in a large Midwestern academic medical center (*n* = 18,000 patients, *n* = 1144 patients meeting age and anthropometric criteria). Additional inclusion criteria included the following: both parent and child were interested in making nutritional and/or physical activity changes, parent with access to smartphone or computer with internet at home (for syncing fitness tracker data), and English-speaking. Parents were required to read and write English. Participants were excluded from the six-session intervention if the child had been seen in our institution’s comprehensive weight management clinic within the last 2 years as we wanted to test the acceptability and satisfaction of the intervention within a population who was naïve to comprehensive weight management counseling. Participants were also excluded or withdrawn if the child had significant behavioral or developmental problems that would impair their participation or disrupt group activities, or if the child was on atypical antipsychotics or chronic steroids, for which weight gain is a common side effect.

Current or past participation in other weight management interventions was not an exclusion criterion for children in the pilot, as we hoped to co-design the sessions with caregivers who had previously been receptive to medical counseling for weight management and share the aspects that were most meaningful for them. Participants from the focus groups were also allowed to participate in the pilot session. Further, we hoped the participants and their parents could share the tips learned from the weight management clinic that they found particularly resonant, while also providing valuable insight about why the comprehensive weight management clinic may or may not have worked for their child.

Patients were recruited for the group sessions through fliers in waiting rooms of the primary care clinic, during their clinic visits, and via mailing letters to potentially eligible families. Focus group members were invited to participate in the pilot session. The full six-session groups were capped at 14 dyads, to allow for dropouts and an anticipated attendance of 80% at each session, with the goal of having approximately 8 dyads at each session, as this number facilitates diverse, but meaningful peer interaction [31]. Letters were sent to participants whose address in the electronic health record (her) indicated they lived in a neighborhood of historic disinvestment. We defined neighborhoods of historic disinvestment as those impacted by redlining practices [15]. These same neighborhoods impacted by redlining have been shown to have disproportionately worse child health outcomes [32]. Recruitment by letter for each cohort was conducted in batches, with families in a single batch living in geographically close zip-codes. This was done to facilitate the sharing of neighborhood-specific tips during sessions. Only one child per family could enroll in the study and was tracked for study measures, but all siblings were invited to join the group sessions. Families expressed interest by contacting the study via phone call. A consent interview was scheduled for eligible dyads prior to the first session.

Participants received $10 for attending the first session. Fitness trackers were distributed at the second session, participants received an additional $10 for the first time they uploaded fitness tracker data, and $10 for the follow-up interview. Payment for attending interim sessions was intentionally not given, as the goal was to assess if these sessions had intrinsic value for participants. A healthy meal, which was eaten family style at the beginning of the sessions, and childcare for when the caregivers and children separated was provided at each session. Parent champions who co-led a group session with a “content expert” were compensated with $40 and recognized with a certificate indicating they had co-facilitated a STEP IN session. All participants who completed the sessions received a certificate of completion.

### 2.4. Setting

Group sessions primarily occurred in a conference room located on the top floor of the same building as the primary medical home. There is free covered parking for the building. For the second half of the session, children went with two study staff members to a nearby conference room for age-appropriate activities (e.g., coloring, simple games) related to the nutrition lesson their caregivers were receiving. The final session in all groups took place in the hospital’s metabolic kitchen where parent–child dyads made their own pizzas and participated in a celebration session.

### 2.5. Study Intervention

After providing consent, parents/caregivers completed a demographic survey questionnaire, the Patient Health Questionnaire, the Perceived Social Support Questionnaire, the Perceived Stress Scale Questionnaire, and Child Intention to be Physically Active [33,34,35,36]. Parent and child height and weight were collected by a trained team member at the consent visit or at the first group session. With input from the focus groups, we conducted an initial pilot session consisting of four every-other-week group sessions and the fitness tracker intervention for parent–child dyads. Pilot participants helped further refine the curricular materials and practical tips to be pragmatic, culturally resonant, and relevant.

Across the three cohorts, each group session consisted of four parts (Figure 1): (1) a healthful meal eaten family style with parents, caregivers, study staff, and siblings; (2) a co-facilitated 20-min discussion for parents only (no children or siblings) of barriers and successes related to healthy eating by a peer and “content expert” [37], (3) a brief interactive teaching session for parents led by the content expert (a medical student, R.R.T., or master of health promotion student, M.E.P.) focused on practical, implementable strategies (e.g., MyPlate, being active, meals on a budget, Table 1), and (4) a review of survey and fitness tracker results for the whole group (parents, children, siblings) with a discussion of successes and problems with the program for participants (Figure 1). Prior to co-facilitating a session, parent champions received a 15-page training toolkit, which covered research ethics, confidentiality, and tips on facilitating and leading discussions. The facilitated discussion operated similar to group therapy, building relatedness and competence among participants as they saw peers facing similar barriers [38,39]. It was hoped that parents would find inherent value in the connection gained through discussion, which would also aid in retention for subsequent sessions. In group sessions, participants collectively agreed to ground rules to be present, turn off cell phones or place on vibrate, and to maintain confidentiality.

During the first session, each caregiver–child dyad received a binder “The STEP IN Playbook” with recipes, some general tips, and a preprinted calendar with upcoming session dates on the cover. Participants were instructed to bring the playbook to each session, as there would be related handouts for each session that went into the binder. Practical tips from the successes that parents shared at each session were recorded in real-time by the primary investigator (K.A.C.) and printed out to be added to their playbook binder at the end of each session. At the end of each session, one parent would volunteer to facilitate the subsequent group session as a parent champion. Each facilitator was given a script one week in advance plus the 15-page facilitator guide with tips on how to guide the discussion and prevent digressions. The idea of a rotating peer co-facilitator was generated from the focus groups.

The sessions strived to foster a group dynamic of “all teach all learn”, recognizing that all participants bring valuable lived experience. We obtained assent to use first names for all participants during group sessions to flatten any perceived hierarchy or power imbalance between the study staff and participants. Although the primary investigator (K.A.C.) was one of several dozen pediatricians in the participants’ primary medical home, she introduced herself by her first name. The primary investigator sat “within the circle” at all group sessions and remained deferential to the content expert and parent champion, gently assisting with facilitation or staying on script only when necessary.

For additional information about the protocol, please see ClinicalTrials.gov registration number NCT02724839 (https://clinicaltrials.gov/ct2/show/NCT02724839?term=NCT02724839&draw=1&rank=1 (accessed on 22 February 2023)).

### 2.6. Study Outcome Measures

Attendance was tracked via a standard written sign-in sheet at the beginning of each session with the research coordinator ensuring accuracy by confirming that every participant signed in. We made up to three attempts to contact caregivers who were lost to follow-up in between the consent visit and initial visit, or after the sessions started. The research coordinator also called or texted each participant who had an unplanned absence to understand the reasoning, with the goal of making the sessions as accessible as possible for future groups. Caregivers self-reported and proxy-reported their child’s prior day consumption of fruits and vegetables, sugar sweetened beverages, fast food, snack food, and TV/screen time via electronic surveys sent to the caregivers’ email from the REDCap database. These same surveys were sent 20 times during the study session and at least 7 times prior to the start of the session to establish a baseline for parent and child habits. Participants were aware the surveys would be repeatedly administered but were unaware of the exact survey distribution dates to promote accuracy and prevent participants from modifying their behaviors just for the survey. The research coordinator texted participants to let them know there was a survey in their email inbox the morning the surveys were sent. A follow-up reminder was scheduled and distributed the same evening if participants failed to submit answers. Surveys of acceptability of the session’s content, using Likert scales, were completed at the end of each session. Additionally, there were baseline and exit surveys consisting of Pediatric Quality of Life (PedsQL Version 4.0), Child Self-Efficacy to be Physically Active, Child Intention to be Physically Active, Parental Perceived Social Support, Parental Perceived Stress, and a screen for parent depressive symptoms, the PHQ9 [33,34,35,36,40]. Parent and child BMI were measured at baseline and after the final session. Surveys were administered via REDCap and response rate was tracked.

The lead investigator (K.A.C.) conducted exit interviews over the phone with caregivers and children who completed, and those who did not complete the study, one to four weeks after their cohorts’ sessions concluded. Participants (both the parent and child) were asked what they liked and disliked about the study, a healthy habit they gained from the study, and what they would change about the study. They were also asked if they were still wearing their fitness tracker, if this study changed their interaction with their doctor/their child’s doctor, and if they were interested in a referral to a subspecialty pediatric weight management program. The purpose of these exit interviews was to capture qualitative feedback on what worked well and what could be improved from the caregivers’ perspective. To the extent possible logistically, these recommendations for improvement were incorporated into the subsequent cohort. As the exit interview was also conducted by a pediatrician in the primary care home, she could offer a referral to caregivers desiring to follow up with the comprehensive weight management clinic.

### 2.7. Statistical Methods and Data Analysis

Percent attendance for each session was calculated and plotted in a time series. Height (measured to the 0.1 cm) and weight (measured to the 0.1 kg) were measured in duplicate on a clinic stadiometer and an electric scale, and a third measure was taken if the two measures were greater than 0.1 cm or 0.1 kg apart from one another. The average of the height and weight measures were used to calculate BMI for parents and BMI z-score for the study children [41]. The purpose of collecting heights and weight was to assess the feasibility of study procedures for a future efficacy trial; this feasibility trial was not powered to measure BMI outcomes. Individual- and group-level data for self-reported consumption outcomes (fruits and vegetables, sugar-sweetened beverages, fast food, snack-food, TV/screen time) were plotted in time series on run charts which were shared with participants each week. Statistical process control was used to evaluate the impact of our interventions. A centerline (mean) and control limits were established using pre-intervention survey data. The mean was recalculated, and the centerline adjusted when the following criteria were met: a “shift” of seven or more consecutive points either above or below the mean or a “trend” of at least five consecutively increasing or decreasing data points. These are considered statistically significant changes, as there is <5% chance of observing these patterns by chance [42]. Acceptability results were calculated using means of responses on a scale of 1 to 5. Qualitative feedback from the focus groups and the exit interviews were compiled and shared with the design team advisory board for comment and input.

## 3. Results

### 3.1. Study Population and Recruitment

Of those eligible for the pilot group, seven parent–child dyads provided consent and enrolled. Of the seven, four were previously part of a focus group. For the biweekly and monthly cohorts, a total of 903 letters were mailed. A total of 87 caregivers expressed interest, however 35 were later found to be ineligible or could not attend for reasons such as transportation, or scheduling conflicts (Figure 2). Most caregivers who expressed interest heard about the study via the letter, a few (*n* = 5) reported seeing the flyers in the clinic. Of the remaining 52 eligible, 27 parent–child dyads were consented and enrolled in the two non-randomized sequential trials occurring between May 2016 and February 2017. Two participants withdrew prior to the start of the intervention.

### 3.2. Demographics

Thirty-seven caregivers of children with overweight/obesity participated in the focus groups. They were 86% female (*n* = 32), 89% Black (*n* = 33). There were 2 grandparents (5%), and the remaining 35 (95%) were parents.

Caregiver demographic information for the pilot, biweekly, and monthly sessions are in Table 2. Mean child age was 10, 9, and 8.5 for the pilot, biweekly, and monthly groups, respectively. In the pilot and biweekly groups, one caregiver was a grandparent, otherwise all caregivers were parents.

Baseline parent BMI of those who completed the biweekly and monthly cohorts (*n* = 20) was 42.9 (standard deviation [SD] 34.6 to 49.9). Baseline child BMI was 24.2 (SD 21.7 to 28.8) with a baseline BMI z-score of 1.97 ± 0.44.

### 3.3. Attendance

Retention and attendance were highest in the initial pilot sessions (100%, 0 dropouts), high in biweekly cohort (83%, 1 dropout) and moderate in the monthly cohort (62.7%, 4 dropouts, Figure 3). Of the five who did not complete the trial, three participants were lost due to schedule changes and loss of transportation. Attendance remained very high for the biweekly session, even though participants were not paid to attend sessions after the first session. On average participants attended 5 of 6 sessions (83%). Although participants in the monthly cohort were highly engaged at sessions, attendance was irregular (33–100% at each session).

### 3.4. Satisfaction and Acceptability

As for acceptability, surveys of the focus group were positive: 77% of focus groups thought planned group nutrition sessions sounded helpful, 60% said they sounded fun, and 83% thought they could learn from them. Focus group participants liked the idea that they would be led by a parent (and 68% said they would be willing to lead at least one session), however they also wanted a “content expert” to co-lead. One focus group participant suggested that we rotate the parent lead; this suggestion was incorporated into the pilot and feasibility trial.

Participants in the monthly and biweekly group sessions reported high overall satisfaction (4.75 out of 5) with the group sessions. They rated the nutritional information to be useful (4.72 out of 5). Participants also enjoyed having the group sessions led by a peer (4.64 out of 5) and felt connected to the other parents (4.63 out of 5).

### 3.5. Sample of Data Sharing

At each session, participants’ data on steps and habits such as fruit/vegetable consumption and screen time were compiled and shared back with them. Results for parents and children would be compiled separately. Below are a few examples of the data shared with dyads (Figure 4).

### 3.6. Qualitative Outcomes

In exit interviews (*n* = 14, 7 via phone and 7 in-person), families reported more engagement in their primary care home after completion of the STEP IN study. A few families mentioned it seemed the primary care clinic “truly cared about them” to offer this program. Most participants said they enjoyed the interaction. Several who did not know each other before became friends and have spent time together outside of the sessions. Many families were receptive to follow up in primary care or a weight-management clinic.

Additionally, almost all participants said they learned about how to properly portion their own meals and their children’s meals, and this has improved their eating habits. Below is an email from one of the initial pilot session parents:


*“I enjoyed these sessions. I wish there were more. I thought I would let you know that since we first weighed in and as of yesterday I have lost 14 lbs. My child has not lost any, but I think this is because I can control what I eat and have more will power. Dad still buys junk food and if [child] is offered he will eat it. I would be very interested in helping the next group if you need help. I just wanted you know I thought it was a great way to get the kids involved and parents together to realize we are not alone in our struggles with getting kids to eat healthy. And to help us reach a goal. We have made friends and hopefully will continue to build each other up. Thanks.”*


Another participant reflected on the importance of setting and achieving goals and mentioned appreciating not feeling like she was alone in her struggles—“it’s not just me”. The camaraderie and group support seemed to be a key aspect to weight loss success among caregivers who lost weight during the trial.

## 4. Discussion

Overall, the STEP IN feasibility study was successful in achieving engagement, high satisfaction scores, and high rates of attendance with a low-income and predominantly non-Hispanic Black population. A strength of this trial’s format was the opportunity for peer support in behavior change. Several parents shared their own stories about their struggles with weight, and their challenges with how to mentor their child and be a role model when they have not achieved their own weight management goals. Participants agreed that other parents are an excellent source of practical knowledge for how to help their children eat healthier. Parents felt that doctors often do not have time to help families during a typical short visit with tasks such as meal planning, but the duration of the group sessions and practical tips from other parents were excellent opportunities for learning. Participants positively reviewed the incentives (money and the fitness tracker). Parents and children had fun competing over step counts and servings of fruits and vegetables (these habits were compiled and shared back to dyads, Figure 4). The children especially had fun when they “beat” the parents in step counts. Parents also added each other as friends on their FitBit app. This minimal intensity, family-focused intervention shows promise for improving child and parent diets and physical activity. This intervention does not address community level factors, such as neighborhood-level inequities driven by structural racism [43,44,45,46]; however, it does provide an opportunity for additional counseling and peer support which can address some barriers to healthy habits faced by our patients and families. The peer connections and support partially address weight stigma, participants felt less alone in their weight struggles [5].

The study was intentionally designed to trial biweekly versus monthly sessions to assess which had better engagement and results. Participants’ schedules (work or school) were more likely to change over the course of six months compared to the three months required for the biweekly sessions. We also found that six months allowed for more chances of life stressors, such as an illness or death in the family, to change families’ priorities. While we found better attendance with biweekly sessions, one confounding factor is the biweekly cohort had many sessions over summer break from school. Sessions appear to be poorly attended at the end of school semesters (May and December) and in the winter months (December and January) due to snow and cold. An additional confounder was session five for the monthly cohort had to be rescheduled due to weather, yielding 50% attendance. Overall, attendance rates compare very favorably with the clinic appointment no-show rate of 25% for all visit types, and no-show rates as high as 67–75% for weight follow up and nutrition-only visits. The high attendance rate is also crucial given the participants in the full six session cohorts were not previously engaged with an intensive behavior and lifestyle program. Our study visits’ high attendance rates and the high level of engagement of participants at the sessions speak to the value participants see in these sessions for gaining knowledge, feeling supported, and improving their child’s health and their own health. The recent clinical practice guidelines from the American Academy of Pediatrics for the treatment of children with obesity recommend intensive behavior and lifestyle treatment as a key intervention for children six years and older, with a minimum of 26 contact hours recommended [47]. The prevalence of obesity necessitates a marked increase in the number and the availability of these programs especially in under-resourced communities where obesity rates are highest and programs most scarce. This feasibility study is a promising first step in developing an intervention to improve healthy nutrition and activity for a high-risk population. The STEP IN intervention could also be considered a stepping stone based in the primary medical home to increase family engagement in intensive behavioral and lifestyle treatment.

This feasibility study has a few important limitations and challenges. First, recruitment was best achieved by letters mailed directly to families (as compared to fliers in the clinic), however there was a low response rate even to letters (about 1%). Optimization of the recruitment strategy will be needed for a larger trial. Finding engaging activities for the children was sometimes challenging and dependent upon the childcare provider’s engagement with the material. An additional challenge was the unpredictable attendance by siblings of a wide age range. Obtaining catering for the shared meal that was healthy, within budget, and enjoyed by the participants was a challenge. While the exact content of STEP IN may need to be customized to other neighborhood or populations, the structure of the peer led sessions could be applied to other groups facing structural disadvantages and high rates of pediatric obesity. We also found that parents with low literacy were not able to participate fully in these sessions, limiting the generalizability of this format of intervention. Finally, the size of the study was not powered to assess the impact of the intervention on objectively measured behavior change or BMI. We did not find a statistically significant change in dyads’ BMI.

Since the study was initially conducted, we have explored models for providing this type of engagement intervention to clinic patients who are in a pre-contemplative or contemplative readiness to change mindset. Because Medicaid does not reimburse for group-level or peer-level counseling in Ohio, and because the program relies on having a full-time program manager to run, we have paused additional study of this intervention. If others are considering group pediatric weight management sessions, we recommend avoiding scheduling the sessions during the end of the academic semester (December and May). Future research should also look for ways to involve children in the learning and sharing their learning while still separating parents and children for part of the session as parents valued this time to connect with one another. Parents noted they would like if their children received their own certificate of completion. We would also consider adding an additional session to discuss sleep hygiene, based on many participants indicating they struggle with healthy sleep habits for themselves and their children.

## 5. Conclusions

This feasibility study of peer-led group sessions showed strong promise and a larger study should be conducted to assess whether the intervention can improve diet and physical activity habits of families who are predominantly low-income and identify as non-Hispanic Black. More frequent sessions (every other week) were found to be better attended than less frequent (monthly). The opportunity for peer support and connection was one key to the success of this low-intensity, family-focused intervention for pediatric weight management based in the primary medical home.

## Figures and Tables

**Figure 1 ijerph-20-05686-f001:**
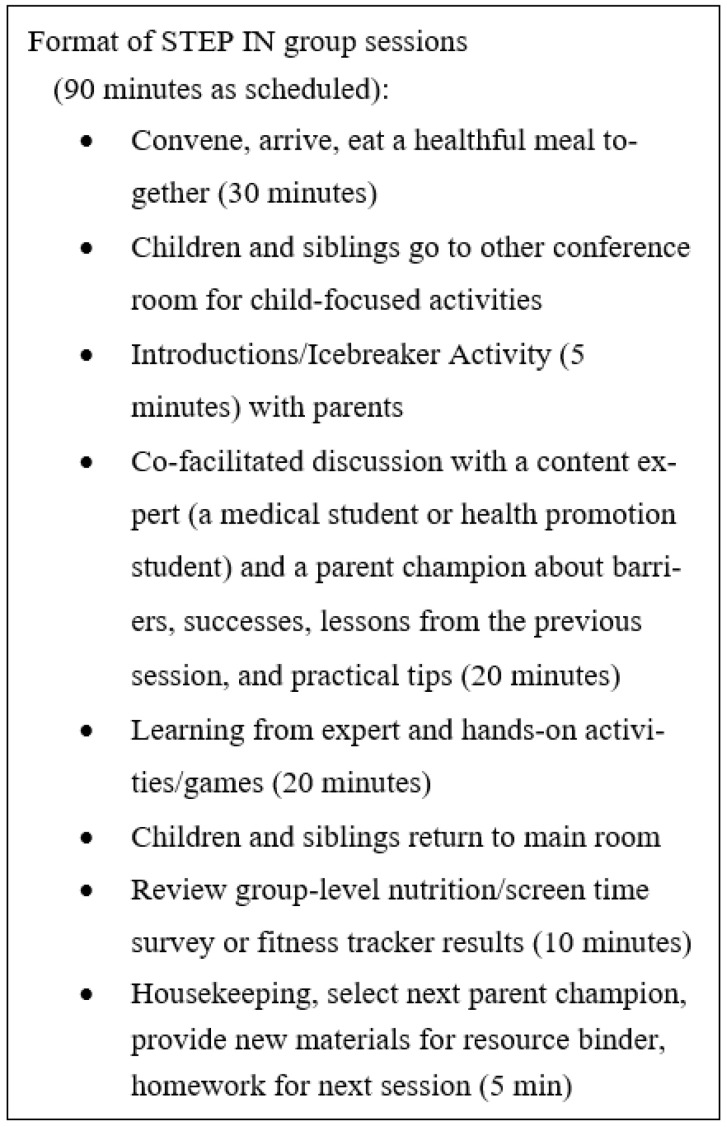
Format of STEP IN sessions.

**Figure 2 ijerph-20-05686-f002:**
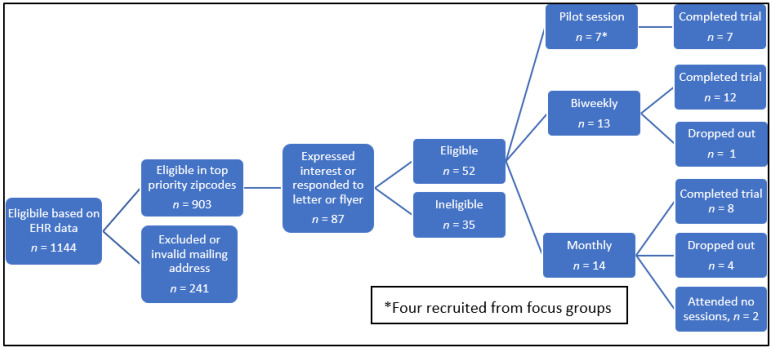
Recruitment flow diagram.EHR: electronic health record.

**Figure 3 ijerph-20-05686-f003:**
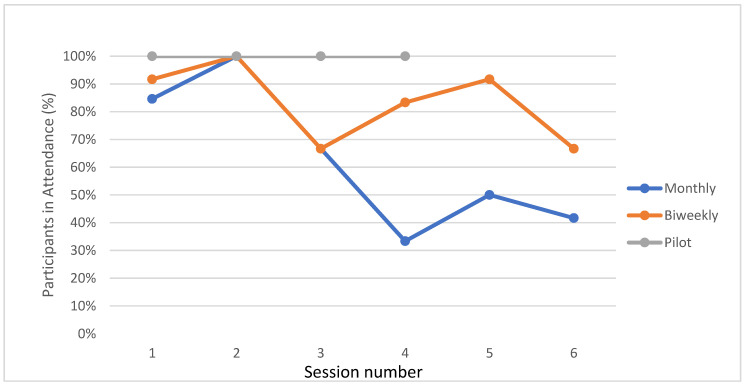
Attendance by session and group. There were 6 sessions in monthly and biweekly groups, only 4 sessions in the pilot.

**Figure 4 ijerph-20-05686-f004:**
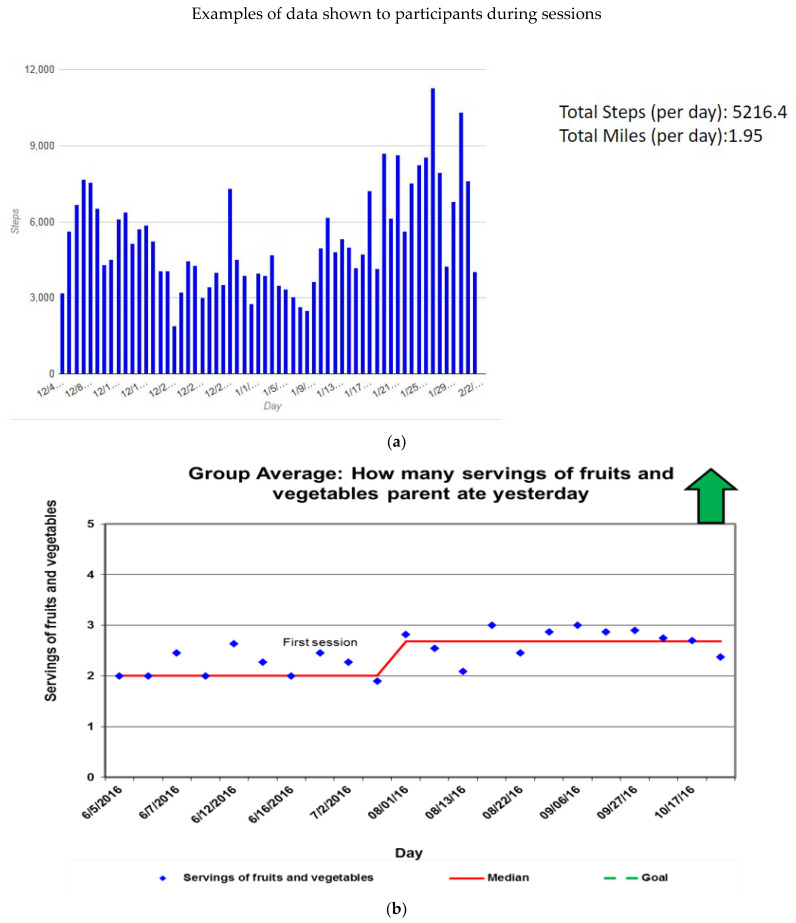
Illustrates three types of data shown to participants during the sessions. (**a**) Shows child daily steps in the monthly cohort; (**b**) illustrates parent reported daily fruit and vegetable consumption, bi-weekly cohort; (**c**) illustrates reported child daily fruit and vegetable consumption in the biweekly cohort.

**Table 1 ijerph-20-05686-t001:** Primary content focus of STEP IN group sessions.

Session Number	Content Focus	Game/Hands-on Activity
1	MyPlate	Jeopardy game about MyPlate and serving sizes
2	I Can Be Active	Fitness trackers administered. True/False game and activity writing down barriers to being physically active on note cards
3	Meals on a Budget	Parents choose meal they most struggle with, create a grocery list, and then swap lists to come up with recipes to make from partner’s list.
4	Working with other Caregivers	Newlywed-style game; parents are paired up and have to guess answer of their partner
5	Healthy Snacks	Iron Chef game to assemble a healthy snack
6	Celebration Kitchen	Build pita pizza together

**Table 2 ijerph-20-05686-t002:** Parent Demographics (*n* = 34 enrolled, *n* = 27 completed).

Characteristics		Pilot *n* (%)	Biweekly, Enrolled *n* (%)	Biweekly, Completed *n* (%)	Monthly, Enrolled *n* (%)	Monthly, Completed *n* (%)
Sex	Female	6 (86)	12 (92)	11 (92)	14 (100)	8 (100)
Race/Ethnicity	Black	4 (57)	11 (85)	10 (83)	10 (71)	6 (75)
	White	3 (43)	2 (15)	2 (17)	1 (7)	1 (13)
	Asian	0 (0)	0 (0)	0 (0)	1 (7)	1 (13)
Age, mean (SD)		40 (14)		35 (5)		36 (8)
Highest Education	HS Diploma	1 (14)		1 (8)		0 (0)
	Some College	5 (71)		8 (67)		5 (63)
	4+ Year	1 (14)		2 (17)		3 (37)

SD: standard deviation.

## Data Availability

The data presented in this study are available on request from the corresponding author. The data are not publicly available due to privacy concerns given the sample size.

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
