# Peer review of "STEP IN: Supporting Together Exercise and Play and Improving Nutrition; a Feasibility Study of Parent-Led Group Sessions and Fitness Trackers to Improve Family Healthy Lifestyle Behaviors in a Low-Income, Predominantly Black Population"

_ijerph, 2023, doi:10.3390/ijerph20095686_

Round 1
Author Response
Thank you for the supportive comments.

Reviewer 2 Report
This was a well-written report of a carefully and appropriately developed and conducted intervention that could help reduce disparities in high weight among lower-income and Black children. The background provides valuable framing of the issue, and I especially appreciate their acknowledgment of upstream and non-modifiable factors that contribute to high weight. The methods were clearly described and appropriate, and the results were similarly clear and supported by the tables and figure. Personally, I would like to see more exploration of healthy lifestyle interventions among young people without an explicit weight loss aim, and I think this intervention is a wonderful model for such a program, although I understand that weight loss was a primary aim of the authors for this intervention.
I have very few suggestions for change, but those that I have are as follows:
Line 43: I am not aware of evidence that obesity directly causes depression and low academic performance, but there is evidence of academic discrimination based on weight (e.g., young people with higher weight receiving lower grades despite no differences in IQ or performance on standardized tests), and weight stigma is widely reported by people with lived experience to affect their mental health, so I think it is more likely that weight stigma is the causal mechanism than high weight itself for these conditions. And, while prospective cohort data do support an association between childhood obesity and premature mortality, the models I've seen do not control for lifestyle factors or the impact of weight stigma, and many do not control for other health confounders such as CVD risk factors. I suggest modifying this sentence to indicate that obesity is *associated with* an increased risk of some health conditions and premature mortality, and perhaps noting that weight stigma can have a powerful negative impact on the lives of people with high weight.
line 150: Add reference to CDC norms for the 85th percentile (assuming CDC norms were used)
line 157: Can you provide a brief description of what you mean by a comprehensive weight management clinic? It wasn't entirely clear what kind of previous weight management programs were allowed vs. disallowed.
Reviewer 3 Report
The manuscript discusses an intervention study designed considering child-parent dyads and group sessions for improving health conditions and nutritional behaviour in a low-income, majorly non-Hispanic Black population. Although the topic of the manuscript is interesting, the authors should pay an additional attention to include sufficient information in the methodology section. It is mentioned in the manuscript that the aim was to control the efficacy of the group sessions on weight control of the subjects through measuring the BMI, but eventually in the results section, the authors neither presented the measurements relevant to BMI nor discussed them in the results section. It is required that the authors present the results and add relevant discussions on the presented figures.
I have the following comments to the authors:
1. In section 2.3, lines 159-170, please also mention and add it to the text if there were any exclusion criteria for parents.
2. In figures 1 and 3, provide captions for the figures underneath the figures. Captions should not be provided inside the figures.
In the last bullet point of figure 1, did the authors intend to mention "10 minutes"? please check and correct.
3. Lines 403-404, the sentence: “this suggestion was incorporated this suggestion into the pilot and feasibility trial.”
This sentence should be restructured. avoid using the word "suggestion "many times.
4. Figures 4a, 4b, and 4c should be presented in one figure as figure 4 and in the caption, the authors should mention what sections a, b, and c, present. Different parts of the same figure should not be separated.
5. Lines 498-499, the sentence: “Finally, the size of the study was not powered to assess the impact of the intervention on objectively measured behaviour change or BMI.”
In this section, a separate figure on changes in BMI of the subjects, i.e. children and parents should be provided. It is required to show the outcome of the study. The results shown in Figure 4 do not highlight the final outcome of controlling obesity in the subjects and the reader is interested to see the main findings of the intervention even if it was not completely satisfactory and the size of the study was not powered as the authors mentioned.
6. Punctuation mark “.” Should be provided after the references throughout the manuscript.
7. Minor grammatical mistakes should be corrected:
· Line 442, “achieve” should change to “achieving”
· Line 507, “involve” should change to “to involve”
8. Keywords should all start with a capital letter.
Reviewer 4 Report
The manuscript describes a feasibility study of a family-centered pediatric weight management intervention. I applaud the authors and study team for their efforts in implementing the program with specific attention to areas of historic disinvestment and historically marginalized populations. Overall, the manuscript is of scientific importance and of interest to readers and I do not see the need for major changes. I do have a few small comments below:
Introduction:
Lines 81- 82 touch on the structural barriers that may be faced by patients, but I believe the introduction could use an additional discussion of structural barriers specific to African Americans and include a discussion of nutrition-related barriers. For example, targeted marking of unhealthful foods to African Americans and the increased likelihood of African Americans living in food swamps, or areas with an increased density of unhealthful food outlets like fast food restaurants. Perhaps a few sentences devoted to these differences could be included, either following line 82 or lines 57-58.
Lines 131-137: I believe more information is needed with regards to the use of user-centered design principals. What types of questions were asked in the focus groups? Perhaps the focus group guide could be included as supplementary material. Additionally, how did the study team arrive at the consensus note documents? Was any process of qualitative coding used? Readers would benefit from additional transparency in this section.
Line 153: I do not believe any changes need to be made to the paper with regards to this sentence, but I wanted to draw attention to this inclusion criteria. It would not need to be discussed in the limitations of this article describing the feasibility of the intervention, but any discussion of evaluating the intervention and changes made as a result of the intervention would need to discuss this inclusion criteria as a limitation of the study- people who are interested in making changes may be more likely to benefit from the intervention.
Line 179: How did you define neighborhood of historic disinvestment? Perhaps a short discussion of the neighborhoods is warranted. Were they affected by redlining, highway construction projects, etc.?
Lines 222-226 and 270-72: I applaud the study team for this structure. Facilitating learning among participants rather than using only a top-down approach to learning is a key component in assessing quality of research that implements qualitative methods, such as your focus groups.
Line 410: The presentation of these data sharing graphs could be expanded. There is significant graphical space in the paper devoted to the examples, but there is no further discussion of the graphs either in the results or discussion section. I understand this is a feasibility study, but perhaps a couple sentences could be added describing the results in the results section and possibly a sentence or two describing participant experiences with data sharing in the discussion section.
Thank you again for allowing me to provide suggestions and for conducting this laudable initiative with a historically marginalized population.
